# Protective impacts of household-based tuberculosis contact tracing are robust across endemic incidence levels and community contact patterns

Joshua Havumaki[1]*, Ted Cohen[1], Chengwei Zhai[2], Joel C. Miller[3], Seth D. Guikema[2], Marisa C. Eisenberg[4], Jon Zelner[5]

1 Department of Epidemiology of Microbial Diseases, Yale University, New Haven, Connecticut, United States of America, 2 Department of Industrial and Operations Engineering, University of Michigan, Ann Arbor, Michigan, United States of America, 3 School of Engineering and Mathematical Sciences, La Trobe University, Bundoora, Australia, 4 Departments of Epidemiology, Mathematics, Complex Systems, University of Michigan, Ann Arbor, Michigan, United States of America, 5 Center for Social Epidemiology and Population Health, Department of Epidemiology, University of Michigan, Ann Arbor, Michigan, United States of America

* joshsh@umich.edu

**Data Availability Statement:** All relevant data are within the manuscript and its Supporting information files.

## Abstract

There is an emerging consensus that achieving global tuberculosis control targets will require more proactive case finding approaches than are currently used in high-incidence settings. Household contact tracing (HHCT), for which households of newly diagnosed cases are actively screened for additional infected individuals is a potentially efficient approach to finding new cases of tuberculosis, however randomized trials assessing the population-level effects of such interventions in settings with sustained community transmission have shown mixed results. One potential explanation for this is that household transmission is responsible for a variable proportion of population-level tuberculosis burden between settings. For example, transmission is more likely to occur in households in settings with a lower tuberculosis burden and where individuals mix preferentially in local areas, compared with settings with higher disease burden and more dispersed mixing. To better understand the relationship between endemic incidence levels, social mixing, and the impact of HHCT, we developed a spatially explicit model of coupled household and community transmission. We found that the impact of HHCT was robust across settings of varied incidence and community contact patterns. In contrast, we found that the effects of community contact tracing interventions were sensitive to community contact patterns. Our results suggest that the protective benefits of HHCT are robust and the benefits of this intervention are likely to be maintained across epidemiological settings.

## Author summary

Screening household members of newly detected tuberculosis cases is an efficient method for finding previously undiagnosed cases in high-burden settings. Despite the intuitive

**Funding:** JH was supported by the NIH National Institute of General Medical Sciences [Grant Number: U01GM110712], TC was supported by NIH National Institute of Allergy and Infectious Disease [R01 AI112438] and JZ was supported by a grant from the Michigan Institute for Computational Science and Discovery (MICDE) at the University of Michigan. The funders had no role in study design, data collection and analysis, decision to publish, or preparation of the manuscript.

**Competing interests:** The authors have declared that no competing interests exist.

appeal of this approach, randomized trials examining the population-level effects of these interventions in settings with sustained community transmission have shown mixed results. One explanation for these inconclusive findings is that household transmission is responsible for a varying proportion of overall tuberculosis burden between locations, with the impact of household transmission being a function of both the overall incidence and the relative intensity of disease-transmitting contacts in the community and the household. In this manuscript, we use an individual-based network model to explore how local incidence levels and patterns of community contact impact the effectiveness of household-based approaches for interrupting tuberculosis transmission. Our analyses suggest that protective benefits of household-based interventions are maintained across a wide range of epidemiological settings. Our findings provide evidence for the robustness of household-based interventions and suggest that variable results from trials may be primarily due to implementation challenges rather than inherent limitations of these interventions.

## Introduction

Despite recent progress in the development of new tuberculosis (TB) diagnostics [1], drugs [2], and vaccines [3], the decline in TB incidence remains far too slow to meet global targets for TB control. We need more effective case detection strategies to extract the maximum benefit available from existing tools. There is an emerging consensus that passive TB case-finding (i.e. waiting for individuals with symptoms consistent with TB to seek medical care), is insufficient and must be augmented with more active approaches that allow cases to be detected and treated as early as possible [4].

Household contact tracing (HHCT) has been advocated as an efficient approach to TB treatment and prevention [5, 6], both because household contacts of known TB cases are at high risk of transmission, and because they are relatively easy to identify. These interventions typically begin when a *household index case* is diagnosed with TB after presenting for care, i.e. ascertained by passive case-finding. Household contacts may then be screened and treated for active TB and latent TB infection (LTBI). HHCT is routinely applied in high-income, low-incidence settings. However, evidence regarding its efficacy in reducing population-level TB risk in higher-incidence settings is mixed. While HHCT has been demonstrated to improve the yield of TB case finding over passive detection [7], conclusions about the impact of HHCT on population-level TB incidence from cluster-randomized trials are varied, with some demonstrating improved TB control at the community level [8] while others failing to show robust population-level effects [9].

One potential explanation for these heterogeneous outcomes is the relative concentration of TB transmission in households and the community. Intuitively, where the fraction of transmission concentrated within households contributes to a greater share of overall TB burden (e.g., in low-incidence settings), we might expect HHCT to be more effective than in settings where there is a greater community burden of TB and household transmission represents a smaller fraction of the total (e.g., high-incidence settings). The average number of contacts and physical distance between contacts (i.e., social mixing patterns) can also affect the distribution of TB transmission in households and the community. For example, longer distances between community contacts will lead to fewer shared contacts between household members. This will likely result in a smaller fraction of co-prevalent household cases and HHCT therefore being less effective. In order to better understand the complex relationship between endemic

incidence and social mixing patterns on the impact of HHCT, we developed a spatially-explicit network model of coupled household and community TB transmission. We then used this model to characterize the relationship between endemic incidence levels and community contact patterns on the impact of HHCT.

## Methods

In this section, we outline the components of our individually-based network transmission model, which extends a previously published individual-based TB model of coupled household and community transmission [10] by introducing spatially structured community contact networks adapted from [11]. Specifically, we extended the Gaussian community contact network from [11] to include household transmission. Further, we extended the natural history model from [10] to include TB transmission across the network and added in separate intervention (e.g., treatment) compartments.

### Spatial contact network

Utilizing a network representation of community contact allows us to assign discrete contacts at the individual level, which will then be used for the contact tracing interventions described below. Contacts represent individuals who interact with each other and are therefore potential tuberculosis transmission links. They may be close contacts (e.g., within households) or casual contacts (e.g., within the community). Our model represents a population consisting of 100,000 individuals divided evenly into 20,000 households (5 individuals per household). Each household is placed uniformly on a 2-dimensional grid. The spatial extent of the grid is implicitly set by the network density, which we fixed to equal 1. Individuals within households are all connected to each other, while community contacts are formed according to a Gaussian connectivity kernel in which the probability of connection between individuals varies as a function of physical distance between their households. On the network level, this kernel controls both the average number of contacts each individual has (average degree) as well as the average physical distance between community contacts (average connection radius). For instance, in networks with a higher average degree, individuals have more potential transmission links from the community. Furthermore, networks with a lower average connection radius have contacts (potential transmission links) that are closer together and may lead to transmission that is more clustered and less widely dispersed throughout the community. For simplicity, we assumed that household and community contacts are static for the duration of the simulation. See S1 Text for more details on the construction of community contact networks.

To explore the robustness of HHCT to different intensities and configurations of community contact (i.e., to examine the impact across different community contact settings), we conducted simulations across a wide range of connectivity kernel parameters (see Table 1). To account for random variation in contact network structure we generated 10 network realizations for each input parameter set. We also calculated the global clustering coefficient, $C$,

**Table 1. Network parameters for connectivity kernel.**

| Parameter | Description | Range | Source |
|---|---|---|---|
| $n$ | Average degree i.e., the total number of community contacts | 25 to 200 | A large range is set to explore variation in community contact. Actual range of generated networks was slightly different. |
| $\sigma$ | Average connection radius | 0.5 to 5 | Range set to obtain predetermined average degree distribution. Actual range of generated networks was slightly different. |
| $\rho$ | Network density: i.e., density of households in grid | Fixed at 1 | Assumption |

(S1 Text) for each network [12] among community contacts, which measures the degree to which individuals have overlapping contacts. Specifically, for a given generated network, the clustering coefficient was calculated by finding all triads (groups of 3 households) with $\geq 2$ connections (i.e., households were deemed 'connected' if there was at least one community contact between them). Then dividing all triads with 3 ties by the total number triads [12]. See Fig 1 for a schematic of household and community network structure and example networks with different clustering coefficients and S1 Fig for features of generated networks.

**Model of TB infection and progression.** The natural history of TB is characterized by a multi-stage latency period, in which recently infected individuals are at highest risk of progression to active disease, but may also enter a state of long-term latency from which they may progress to disease many years after infection, with most infected individuals ($\sim$80%) never

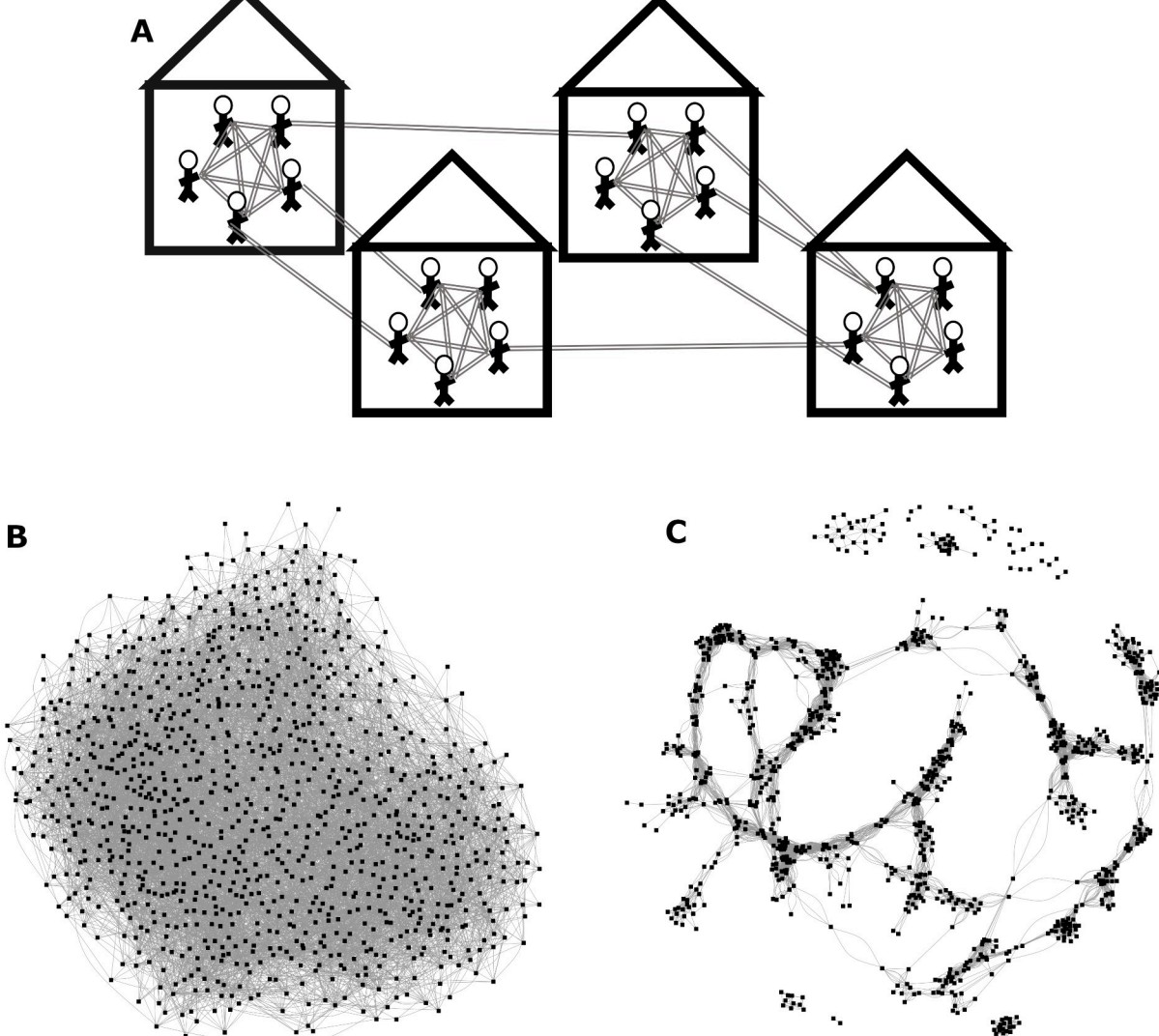

**Fig 1. Network structure.** (A) Schematic of Network Structure (top): All individuals are fully connected within their households. Individuals form community contacts based on a Gaussian (normally distributed) connectivity kernel [11]. Networks consist of 100,000 individuals divided evenly into 20,000 households. (B & C): Example networks consisting of 1000 nodes (i.e., households) and similar average degrees with minimal (B; bottom left) and high levels (C; bottom right) of community clustering.

developing active TB. See S2 Fig for the distribution of latent progression times and the fraction progressing to active TB across different parameterizations of the model. Because early detection is key to the success of HHCT, our model must provide an adequate representation of these transitions. To accomplish this, we adapted a previously published model (from [10]) which includes 5 key disease states. Individuals are born as uninfected and susceptible (*S*). Upon infection, individuals enter an early latent period (*EL*) in which the annual rate of progression to infectious active TB (*I*) decreases over five years before entering the late or long-term latent (*LL*) state from which a small subset of individuals may progress to active TB. Infectious individuals may then enter the recovered state (*R*) spontaneously or as a function of treatment. Finally, individuals may exit the model by death as a function of TB mortality or the background mortality rate. When estimating the rate at which individuals become infected i.e., the force of infection (FOI) on individual *i* in household *j* at time *t* ($\lambda_{ij}(t)$), we account for variable intensities of household and community contact as follows:

$$\lambda_{ij}(t) \quad = [\beta_{HH}I_j(t) + \beta_C I_i(t)](1 - \omega(z_i(t)) \tag{1}$$

Where $\beta_{HH}$ and $\beta_C$ are the *per-contact* transmission rates of household and community contacts, and $I_j(t)$ and $I_i(t)$, are the number of infectious contacts in household *j* and in individual *i*'s community contact network at time *t*, respectively. Finally, because prior infections confer limited protective immunity [13, 14], the FOI is scaled by $\omega(z_i(t)) \in [0, 1]$, where $z_i(t)$ denotes the state of individual *i* at the time of exposure. Fig 2 illustrates the disease states and transitions represented in our model. For a detailed description of the model see S1 Text.

**Model parameters.** Where possible, we obtained point estimates and uncertainty intervals for natural history parameters from published sources, including reviews of historical studies conducted prior to the advent of TB chemotherapy, which provide information on rates of death and spontaneous recovery from untreated active TB [15]. We also consulted systematic reviews e.g., for the rate of progression from *LL* to active TB [16], and modeling analyses e.g., for the rate of progression from *EL* to active TB [17]. Household and community transmission rates were calibrated to reproduce a range of incidence levels (20-400 cases per 100,000 person-years). We chose this range to focus on epidemiological settings in which there was

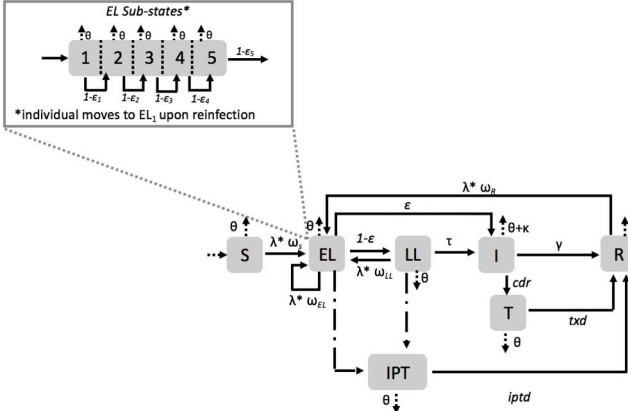

**Fig 2. TB transmission model.** Schematic of TB Transmission Model. Where *S* is susceptible; *EL* is early latent (divided in sub-states to represent the decreased rate of progression to *I* with time since infection); *LL* is late latent; *I* is infectious active TB; *R* is recovered; *T* is treatment; *IPT* represents individuals who are currently taking preventive therapy. Individuals transitioning to the treated states (*T*, and *IPT*) are represented by dot-dashed lines. The rate at which individuals transition to the *IPT* states depends on the number of contacts. Births and deaths are represented by dotted lines.

sustained endemic community transmission. Furthermore, in the most countries, TB incidence levels are within this range. For instance, Bulgaria has an estimated annual incidence of 22 per 100,000 person-years, Peru has an estimated annual incidence of 123 per 100,000 person-years, and Eswatini has an estimated annual incidence of 329 per 100,000 person-years [18]. Following results indicating a higher per-contact rate of transmission from household vs. community contacts [19], in all simulations, we constrained the per-contact community transmission rate to be less than or at most equal to the household transmission rate, i.e. $\beta_C \leq \beta_{HH}$. However, since individuals are likely to have more community than household contacts, this does not exclude the possibility that an infectious individual will cause substantially more infections in the community than their household. Because we were primarily interested in examining how the relative proportions of TB transmission occurring in households and the community alters the impact of interventions, we allowed for a large range of household and community transmission rate values and constrained other key drivers of incidence e.g., the rate at which individuals with active TB seek care. See Table 2 for a full list of transmission model parameters and their sources.

**Passive case finding.** In all simulations, household index cases were ascertained via passive case finding [24, 26], representing current practice in most TB endemic settings [27]. We assumed that on average, it takes one year for an individual with active TB to be detected by

**Table 2. Parameter values and uncertainty ranges.**

| Parameter | Description (units) | Range | Source/Explanation |
|---|---|---|---|
| Natural History | | | |
| $\beta_{HH}$ | Household transmission parameter | 0 to 1.25 | Upper bound is set empirically based on which $\beta_{HH}$ value when equal to $\beta_{uC}$ results in the target incidence levels (20 to 400 cases per 100,000 person-years) in a sufficient number of simulations. |
| $\beta_{uC}$ | Unscaled community transmission parameter | 0 to $\beta_{HH}$ (i.e., $\beta_{uC} \leq \beta_{HH}$) | Derived by multiplying $\beta_{HH}$ by a scaling factor between 0 and 1. $\beta_{uC}$ is then divided by the average degree of community contacts ($n-4$; see Table 1) to obtain $\beta_C$ which is used in the FOI. This parameter also has strong impact on overall TB incidence. (See Eq 1) |
| $\omega$ | Amount of immunity conferred by current state (%) | For $S$ = 0%; $EL$, $LL$, and $R$ = 80%; $I$, $T$, $IPT$ = 100% | [13, 14] |
| $\theta$ | Life expectancy (years) | 72.38 | Global average life expectancy. [20]. The mortality rate used in the model is the inverse of the life expectancy |
| $\epsilon$ | Early latency progression rates (/yr) | A value between 0.0817 to 0.0905 is sampled and then multiplied by (1, 0.41, 0.13, 0.086, 0.028) to derive the rate of progression for each $EL$ sub-state | [17] |
| $\tau$ | Late latency progression (/yr) | 0.0005 | [16] |
| $\gamma$ | Recovery rate (/yr) | 0.09 to 0.15 | [10, 15] |
| $\kappa$ | Active TB mortality rate (/yr) | 0.05 to 0.4 | [10, 15] |
| TB Screening and Treatment | | | |
| cdr | The rate at which individuals with active TB self-present for care (/yr) | 1 | Individuals are detected an average of 1 year after progressing to active TB. We assumed: *Prevalence = Incidence ∗ Duration* and used global numbers from [21] to obtain a disease duration of ∼ 1 year. This is consistent with other analyses e.g., [22, 23]. |
| txd | Treatment duration (months) | 6 | [24] |
| iptd | Preventive therapy duration (months) | 6 | [25] |

this mechanism. Individuals with active TB who are found through passive case finding are given treatment, assumed to no longer be infectious, and eventually recover. In our intervention simulations, passive detection will then trigger the active case finding (ACF) interventions outlined below.

**Household contact tracing.**   For each case discovered through passive case finding, all 4 household contacts of the index case were screened (during the same one-month time step) for active TB and LTBI. This scenario was adapted from [10, 28]. Contacts found to have active TB were placed on treatment, while those with LTBI (*EL* or *LL*) were given preventive therapy. While receiving preventive therapy, we assumed that individuals cannot become reinfected or progress to active TB; at the end of the treatment period, individuals enter the *R* state.

**Alternative active case finding interventions.**   To determine whether any effects of HHCT could be attributed specifically to focusing ACF at the household level, we simulated two additional interventions in which the same number of individuals were screened per detected case: (1) community contact tracing (community CT) in which 4 community contacts of the index case were screened and (2) community-wide ACF, in which 4 randomly selected individuals were screened, following [10, 28]. See S1 Text for additional details of these alternative ACF interventions.

## Simulation strategy

To account for parameter uncertainty and explore parameter values of interest (e.g. $\beta_C$), we ran the model with 9,000 parameter sets obtained using Latin Hypercube Sampling [29] from predefined ranges based either on published estimates or ranges set based on our interpretation of the literature. At the beginning of each simulation run, we selected a network at random and ran the transmission model 5 times using different random number seeds to allow stochastic variation across realizations. We ran the model with passive case finding only until it reached endemic equilibrium. Next, we implemented a simulated ACF trial for 5 years representing a plausible time horizon to evaluate the impact of screening interventions. We ran all interventions and a passive case finding only scenario with each parameter set and random number seed combination. For more details see S1 Text. We generated networks for the model using R version 3.4.1 [30]. Our individual-based network model was run on Matlab version R2019a [31]. We ran our simulations in parallel on a high performance computing cluster and each individual simulation run took between 2 and 10 minutes and used ∼4.5 gigabytes. Model code can be accessed at https://github.com/jhavumaki/network_tb_ibm.

**Measuring the impact of HHCT.**   To assess the impact of HHCT, we calculated the 5-year cumulative incidence rate at the end of the ACF trial period. Incidence was defined as the number of new active TB cases per 100,000 person-years over the previous 60 one-month time steps. We then calculated rate ratios (RRs) to quantify the protective benefit of HHCT for each parameter set and random number seed combination. Specifically, we divided the incidence rate (at the end of the ACF trial period after HHCT) by the passive case finding only scenario incidence rate at the end of the simulation (see S3 Fig for overview of the simulation workflow). Both comparator interventions were also compared to the passive case finding only scenario.

RRs were compared across all parameter sets and also, within strata of network parameters (average degree and average connection radius), and initial incidence-levels (i.e., immediately before the active screening interventions were implemented) to measure the protective benefit conferred by HHCT in different transmission and community contact (i.e., network) settings.

## Results

### Transmission dynamics in the absence of ACF

To ensure that our model behaved as expected and to evaluate the impact of variation in community contact patterns on transmission dynamics, we examined all model runs from the passive case finding only scenario (without restricting incidence to our target range). The proportion of individuals with LTBI was approximately as expected for different TB prevalence levels (S4 Fig). Additionally, a greater proportion of infections were caused by community transmission in higher incidence settings compared with lower incidence settings (S5 Fig). Furthermore on clustered networks, community transmission contributed more to overall TB burden than on less clustered networks. Despite the fact that higher TB incidence levels were driven by more community transmission and that community transmission contributed more to overall TB burden among clustered networks, we observed that as the network clustering coefficient increased, incidence levels were generally lower (S6 Fig). This is explained by the fact that more overlapping contacts (on clustered networks) likely created local contact saturation and depletion of the pool of susceptible individuals [32]. The remainder of the analysis was conducted among simulations within our target range of incidence levels i.e., between 20 to 400 cases per 100,000 person-years.

### Protective benefits of HHCT in different epidemiologic settings

Overall, the protective benefits conferred by HHCT were robust across all settings (i.e., varying incidence level, average degree, average connection radius, and clustering coefficient) with median RRs equaling ∼0.7. See Fig 3 and S13 Fig and S1–S4 Tables for more details.

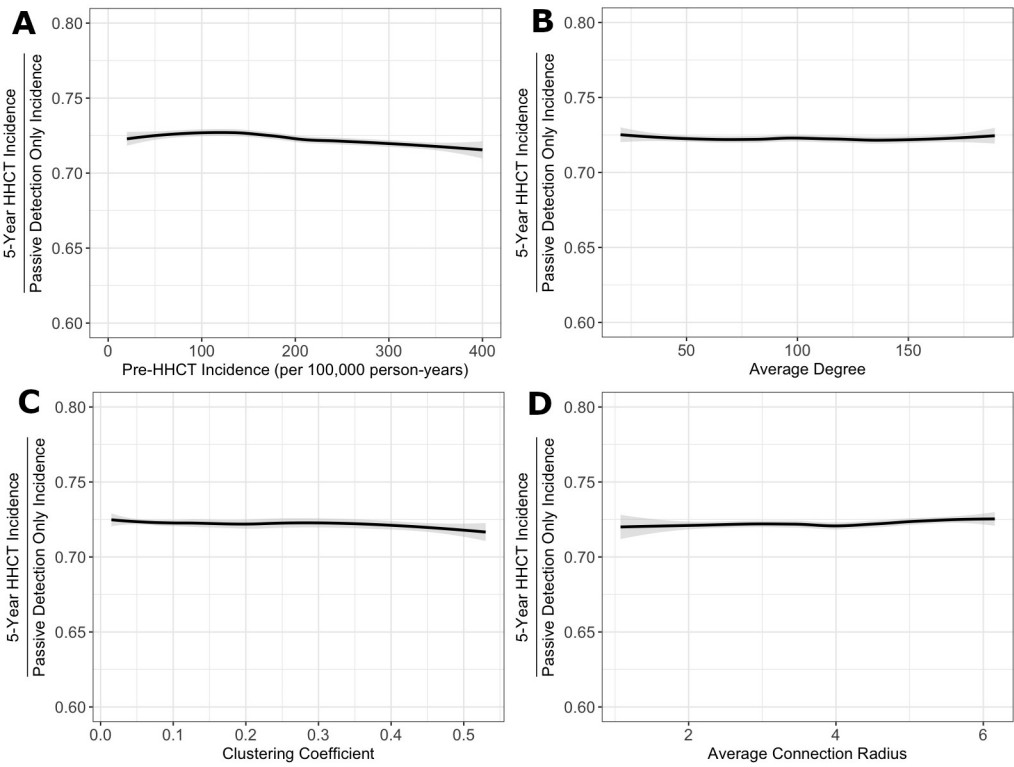

**Fig 3. Effectiveness of HHCT across different settings.** Fitted splines representing relationship between all RRs and (A) the incidence rate immediately before HHCT (per 100,000 person-years) (top left), (B) the average degree (top right), (C) the community clustering coefficient (bottom left), and (D) the average connection radius (bottom right). Lines are splines calculated using the LOESS method in R [33]. Among model runs with incidence rates between 20 and 400 cases per 100,000 person-years. Shaded regions represent 95% confidence intervals.

To ensure that the impact of HHCT was not sensitive to input parameter combinations and endemic incidence levels (e.g., to account for effect modification), we plotted the density of RR values within all pairwise distributions of incidence and network parameter strata (e.g., incidence level by average degree, or average degree by average connection radius). This enabled us to determine whether e.g., average degree affects the impact of HHCT for a given incidence level (S8–S12 Figs). Overall, there did not appear to be any additional emergent trends. See for details.

## Comparator interventions

Contrary to the robustness of HHCT across settings, community CT was sensitive to average degree, average connection radius, and clustering coefficient with the effect of incidence level being similar to HHCT. For example, as average degree increased, community CT became less effective (Fig 4). Notably although community CT was sensitive to network parameters, standard deviations associated with the RRs were greater than the differences between strata. See S14 Fig and S5–S8 Table for all results.

With respect to community-wide ACF, its effects were close to null and it was most sensitive to incidence level. See S15 Fig and S9–S12 Tables for results.

We made additional comparisons between interventions to further assess the performance of HHCT. First, we examined the infectious period duration for each intervention and found that the median infectious period was ~1 month shorter for HHCT compared with community CT, community-wide ACF and passive surveillance only scenarios (S16 Fig). Next, we estimated the number of secondary cases averted among household contacts due to preventive therapy. As expected, the modeled HHCT intervention prevented more cases than the modeled community CT which in turn, prevented more cases than the modeled community-wide ACF (S17 Fig). We then examined the relative number of preventive therapy administrations compared with treatment administrations for each screening scenario. HHCT led to substantially more preventive therapy administrations compared with the other screening scenarios indicating that it may be a more effective approach to identify and treat individuals with recent

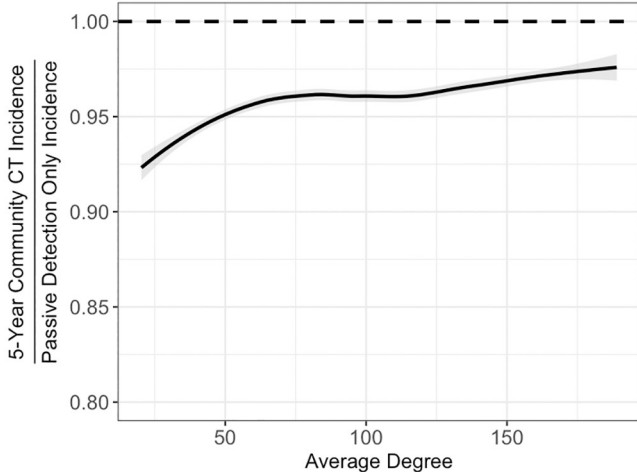

**Fig 4. Effectiveness of community CT varying average degree.** Fitted splines representing relationship between all community CT RRs and the average degree. Lines are splines calculated using the LOESS method in R [33]. Among model runs with incidence rates between 20 and 400 cases per 100,000 person-years. Shaded regions represent 95% confidence intervals.

infection at highest risk of progression (S18 Fig). Finally, we examined the prevalence of LTBI and active TB among household and community contacts. We found that HHCT resulted in lower prevalence of active TB among both household and community contacts, but the prevalence of LTBI was similar between interventions (S19 Fig). The prevalence of LTBI was similar because it represents a cumulative lifetime exposure status and therefore changes substantially more slowly than the prevalence of active TB. Therefore, because we only ran the intervention for 5 years, we did not see major changes in the LTBI prevalence.

## Sensitivity analyses

We conducted additional sensitivity analyses to investigate the extent to which different sources of variability affected the estimated screening scenario impacts. To do this we examined the relative contributions of stochasticity in the disease transmission model, using results from multiple network realizations, and variability in parameter inputs. We constructed a hierarchical regression model to examine the relative contributions of these inputs to variation in estimated RRs across simulations (S13 Table). We found that stochasticity and network realization had only modest effects on RR values, contributing to ~5% and ~3.5% of variability in RRs, respectively. Finally, the overall model $R^2$ was ~85.6% indicating that the parameterization of the model contributed substantially to overall changes in RRs.

Next, we incorporated imported TB cases into the force of infection to understand if a low background level of risk for the entire population might affect the projected impact of interventions. Results from this analysis revealed that although the impacts of all screening interventions were reduced, the main conclusions from our analysis did not change (S22–S24 Figs).

Finally, to explore whether variation in the implementation of HHCT might lead to less robust results, we conducted an additional sensitivity analysis varying the coverage of HHCT. We found that in higher incidence settings (>100 cases per 100,000 person-years), lower HHCT coverage levels resulted in much smaller population-level benefits (S21 Fig).

## Discussion

Our analyses suggest that the protective benefits of HHCT are likely to be robust across diverse settings characterized by variation in incidence level and community contact patterns (Fig 3). As compared to the consistent effectiveness of HHCT, community CT appeared to be more sensitive to different network parameters, suggesting that its utility is more limited to specific scenarios. For instance, as average degree increased, the protective benefits of community CT decreased (S14 Fig). Overall, this suggests that HHCT is robust to different relative proportions of TB transmission occurring in households and the community while community CT is not.

Despite the relative effectiveness of HHCT in reducing community incidence, community transmission was the dominant mode of infection in all but the lowest-incidence settings (see Table 2 and S5 and S7 Figs). This can be partially explained by contact saturation within households [32]. Additionally, transmission causing more infections in higher-incidence settings is consistent with molecular epidemiology studies which have revealed that, in these settings, the majority of co-prevalent TB cases within households are genetically discordant [34, 35]. Thus, both the transmission rates and the number of contacts are important drivers of TB transmission. One might expect HHCT to appear less effective in settings where most TB transmission occurs in the community. However, our results challenge this intuition, as we found that the benefits of HHCT were maintained despite differences in the proportion of transmission events occurring in the community (S20 Fig). This finding is consistent with a

recent model-based analysis indicating that the effects of HHCT are robust to observed variation in local community-level exposure [28].

Our finding of the robustness of the projected benefits of HHCT across settings suggests that differences in community and household forces of infection probably do not explain mixed results of randomized control trials of HHCT to reduce TB incidence [8, 9]. Others have suggested that HHCT may have limited effects in high transmission settings where evidence suggests that a minority of transmission events occur within the home [34–36]. Our results reveal that HHCT has a substantial impact even when within-home transmission does not account for the majority of infection events because household contacts represent a higher-risk population given the likelihood of clustering of exposure risk. This suggests that other explanations may be needed for the mixed results of trials of HHCT, including the coverage and quality of case finding activities. Our sensitivity analysis examining the impacts of varying HHCT coverage confirms that this might be a potential explanation (S21 Fig)

Our approach has some limitations that should be considered in interpreting these findings. For the sake of parsimony, we did not include additional factors in our model that could have altered the risk of TB at the individual level. For example, TB can cluster due to shared risk factors (aside from transmission) like alcohol use and malnutrition [37]. Future models exploring the impact of individual-level heterogeneity on the robustness of HHCT will be helpful in clarifying the real-world utility of this approach. Next, our model does not include an interaction between the intensity of symptoms and the infectiousness of cases, as has been done in earlier models [38]. Including these dynamics would have likely affected the relative performance of all interventions in the same way so it would have not changed our overall conclusions regarding the efficacy of HHCT. At the same time, models including the time-dynamics of infectiousness will be extremely helpful for guiding the timing of the proposed interventions.

It is important to note that the implementation of contact patterns were simplified to reduce the computational demands of our model. First, we assumed static network connections. Dynamic network connections would have allowed infection to escape more easily from local clusters [39], dampening differences across more and less clustered networks. To ensure that we accounted for this, we conducted our analyses across different networks (i.e., with different clustering coefficients) and this likely accounts a wide range of distributions of TB. We also assumed that number of individuals per household is fixed (i.e., at 5 people). Although this is unrealistic, keeping the household size fixed allowed us examine how variation in community contact alone might impact the effectiveness of different screening interventions. Examining the impact of variation in two dimensions (i.e., on both the household and community levels), may lead to different effects. Finally, our births reshuffling scheme based on [39] is not realistic, however, it allows for the population size to be constant and prevents our model from placing susceptible individuals into areas of high endemic transmission.

These results suggest important future directions this research can take to understand the real-world applicability of HHCT in high-burden settings. Our model focused on the effects of heterogeneity across networks with varying average degrees and average connection radii but relatively low within-network variability in connectivity. Future work should consider the effects of heterogeneity within a single network in more detail to capture the impact of variable household sizes, variable average degrees, super-spreading individuals, spatial hotspots, and other causes of right-skewed community contact distributions. Adding within network heterogeneity will alter the local risk of TB. Although the effects of HHCT have been found to be robust to observed variation in local community-level exposure in one setting [28], it is unclear what a systematic exploration of different within network heterogeneities would reveal. Additionally, mechanisms that impact susceptibility and the risk of progression to active TB (e.g., HIV, malnutrition, diabetes) may also lead to more community-level heterogeneity.

## Supporting information

**S1 Fig. Features of generated networks.** Community Average Degree by average connection radius colored by community global clustering coefficient. We generated a wide array of networks i.e., long range connections vs. short range clustered connections and/or many community contacts vs. few community contacts. The parameters we used to specify the networks (see Table 1) were different from the actual calculated metrics on the generated networks. The range of average degrees of generated networks was ~20 to ~190. The range of average connection radii of generated networks was from ~0.8 to ~6.2.
(TIFF)

**S2 Fig. Distributions of latent TB progression times and probabilities.** For all parameter sets, fraction progressing among those in the early latent state and across all latent TB states (left) and the time to progression for individuals in the early latent state and across all latent TB states (right).
(TIF)

**S3 Fig. Workflow of analysis.**
(TIFF)

**S4 Fig. Active TB prevalence per 100,000 individuals vs. latent TB prevalence.** Active TB prevalence per 100,000 individuals vs. latent TB prevalence across all model runs in which no intervention (passive-detection only) was administered. Points correspond to monthly average values over the final year of the simulation. Latent levels were calculated by summing early latent and late latent states.
(TIFF)

**S5 Fig. Infections attributable to community household transmission by incidence level.** Annual number of TB infections (i.e., new cases of *EL*) attributed to community vs. household transmission (y-axis) and incidence levels immediately before ACF (x-axis). Points are colored by community clustering coefficient. These results are excluding model runs that did not result in an outbreak. The line is the fitted spline calculated using the LOESS method in R [33].
(TIFF)

**S6 Fig. Active TB incidence vs. network clustering coefficient.** Active TB Incidence per 100,000 person-years vs. clustering coefficient. These results are excluding model runs that did not result in an outbreak. The line is the fitted spline calculated using the LOESS method in R [33].
(TIFF)

**S7 Fig. Transmission parameters vs. network parameters.** Community beta vs. average degree colored by incidence (top left). Community beta vs. average connection radius colored by incidence (top right). Household beta vs. average degree colored by incidence (bottom left). Household beta vs. average connection radius colored by incidence (bottom right). These results are excluding model runs that did not result in an outbreak.
(TIF)

**S8 Fig. Performance of HHCT across incidence levels.** Ridgeline plot showing performance of HHCT across different pre-ACF incidence levels (per 100,000 person years) using passive detection only as a reference group. The median is denoted by the solid vertical black line and the dashed vertical black line denotes a null rate ratio (RR) equal to 1.
(TIFF)

**S9 Fig. Performance of HHCT across incidence level and average degree.** Ridgeline plot showing performance of screening interventions within strata of average degree and incidence (per 100,000 person years). The median is denoted by the solid vertical black line. The dashed vertical black line denotes a null RR equal to 1.
(TIFF)

**S10 Fig. Performance of HHCT across incidence level and average connection radius.** Ridgeline plot showing performance of screening interventions within strata of average connection radius ($\sigma$) and incidence (per 100,000 person years). The median is denoted by the solid vertical black line. The dashed vertical black line denotes a null RR equal to 1.
(TIFF)

**S11 Fig. Performance of HHCT across average degree and average connection radius strata.** Ridgeline plot showing performance of screening interventions within strata of average degree and average connection radius ($\sigma$). The median is denoted by the solid vertical black line. The dashed vertical black line denotes a null RR equal to 1.
(TIFF)

**S12 Fig. Performance of HHCT across incidence level and clustering coefficient.** Ridgeline plot showing performance of screening interventions within strata of community clustering coefficient and incidence level. The median is denoted by the solid vertical black line. The dashed vertical black line denotes a null RR equal to 1.
(TIFF)

**S13 Fig. 100 best performing model runs.** The 100 best performing model runs (i.e., with the lowest rate ratios), for HHCT and among incidence rates between 100 to 200 cases per 100,000 person years. We plotted these trajectories immediately before and after ACF was implemented. The vertical dashed red line indicates the time step in which active screening interventions were implemented. The line is the fitted spline calculated using the LOESS method in R [33].
(TIFF)

**S14 Fig. Performance of community CT across settings.** Fitted splines representing relationship between all RRs comparing community CT to passive surveillance only and (1) the incidence rate immediately before community CT (per 100,000 person-years) (top left), (2) the average degree (top right), (3) the community clustering coefficient (bottom left), and (4) the average connection radius (bottom right). Lines are splines calculated using the LOESS method in R [33]. Among model runs with incidence rates between 20 and 400 cases per 100,000 person-years. Shaded regions represent 95% confidence intervals.
(TIF)

**S15 Fig. Performance of community-wide ACF across settings.** Fitted splines representing relationship between all RRs comparing community-wide ACF to passive surveillance only and (1) the incidence rate immediately before community-wide ACF (per 100,000 person-years) (top left), (2) the average degree (top right), (3) the community clustering coefficient (bottom left), and (4) the average connection radius (bottom right). Lines are splines calculated using the LOESS method in R [33]. Among model runs with incidence rates between 20 and 400 cases per 100,000 person-years. Shaded regions represent 95% confidence intervals.
(TIF)

**S16 Fig. Average infectious period duration.** Ridgeline plot showing how the average infectious period in months varies by screening intervention. The median is denoted by the solid

vertical black line.
(TIFF)

**S17 Fig. Average secondary cases averted from preventive therapy.** Ridgeline plot showing how the average number of secondary cases averted among household contacts varies by screening intervention. The median is denoted by the solid vertical black line.
(TIFF)

**S18 Fig. Ratio of preventive therapy to treatment by interventions.** Ridgeline plot showing the total number of preventive therapy administrations divided by the total number of treatment administrations by screening intervention. The median is denoted by the solid vertical black line.
(TIFF)

**S19 Fig. Prevalence of LTBI and active TB among household and community contacts.** Ridegeline plots representing the prevalence of LTBI and active TB among household and community contacts across all simulation runs. (1) LTBI among household contacts (top left), (2) Active TB among household contacts (top right), (3) LTBI among community contacts (bottom left), and (4) Active TB among community contacts (bottom right). The median is denoted by the solid vertical black line.
(TIF)

**S20 Fig. Household vs. Community transmission and HHCT performance.** Fitted splines representing relationship between all RRs comparing HHCT to passive surveillance only and the number of community attributable infections to household attributable infections. We removed extreme community to household transmission ratios <2.5% and >97.5% to make the figure easier to interpret. Lines are splines calculated using the LOESS method in R [33]. Among model runs with incidence rates between 20 and 400 cases per 100,000 person-years. Shaded regions represent 95% confidence intervals.
(TIFF)

**S21 Fig. Varying HHCT coverage.** Ridgeline plot showing performance of screening interventions within strata of HHCT coverage and incidence level. The median is denoted by the solid vertical black line. The dashed vertical black line denotes a null RR equal to 1.
(TIFF)

**S22 Fig. Performance of HHCT across settings with Imported TB Cases.** Fitted splines representing relationship between all RRs comparing HHCT to passive surveillance only and (1) the incidence rate immediately before HHCT (per 100,000 person-years) (top left), (2) the average degree (top right), (3) the community clustering coefficient (bottom left), and (4) the average connection radius (bottom right). Lines are splines calculated using the LOESS method in R [33]. Among model runs with incidence rates between 20 and 400 cases per 100,000 person-years. Shaded regions represent 95% confidence intervals.
(TIF)

**S23 Fig. Performance of community CT across settings with Imported TB Cases.** Fitted splines representing relationship between all RRs comparing Community CT to passive surveillance only and (1) the incidence rate immediately before community CT (per 100,000 person-years) (top left), (2) the average degree (top right), (3) the community clustering coefficient (bottom left), and (4) the average connection radius (bottom right). Lines are splines calculated using the LOESS method in R [33]. Among model runs with incidence rates between 20 and 400 cases per 100,000 person-years. Shaded regions represent 95%

confidence intervals.
(TIF)

**S24 Fig. Performance of community-wide ACF across settings with imported TB cases.** Fitted splines representing relationship between all RRs comparing Community-wide ACF to passive surveillance only and (1) the incidence rate immediately before Community-wide ACF (per 100,000 person-years) (top left), (2) the average degree (top right), (3) the community clustering coefficient (bottom left), and (4) the average connection radius (bottom right). Lines are splines calculated using the LOESS method in R [33]. Among model runs with incidence rates between 20 and 400 cases per 100,000 person-years. Shaded regions represent 95% confidence intervals.
(TIF)

**S1 Table. HHCT RRs by incidence strata in order of performance.**
(PDF)

**S2 Table. HHCT RRs by average degree strata in order of performance.**
(PDF)

**S3 Table. HHCT RRs by average connection radius in order of performance.**
(PDF)

**S4 Table. HHCT RRs by clustering coefficient strata in order of performance.**
(PDF)

**S5 Table. Community CT compared with passive surveillance only RRs by incidence strata in order of performance.**
(PDF)

**S6 Table. Community CT compared with passive surveillance only RRs by average degree strata in order of performance.**
(PDF)

**S7 Table. Community CT compared with passive surveillance only RRs by average connection radius in order of performance.**
(PDF)

**S8 Table. Community CT compared with passive surveillance only RRs by clustering coefficient strata in order of performance.**
(PDF)

**S9 Table. Community-wide ACF compared with passive surveillance only RRs by incidence strata in order of performance.**
(PDF)

**S10 Table. Community-wide ACF compared with passive surveillance only RRs by average degree strata in order of performance.**
(PDF)

**S11 Table. Community-wide ACFcompared with passive surveillance only RRs by average connection radius in order of performance.**
(PDF)

**S12 Table. Community-wide ACF compared with passive surveillance only RRs by clustering coefficient strata in order of performance.**
(PDF)

**S13 Table. Hierarchical model.** Exploring the impacts of stochasticity, network realization and model parameterization on RRs.
(PDF)

**S1 Text. Additional details about methods, and results.** Network kernel formula, generated network features, more details about natural history model and interventions, simulation workflow, epidemiology of modeled population, performance of screening interventions within strata of network parameters and incidence, model trajectories of best performing runs.
(PDF)

## Author Contributions

**Conceptualization:** Joshua Havumaki, Ted Cohen, Jon Zelner.

**Data curation:** Joshua Havumaki, Jon Zelner.

**Formal analysis:** Joshua Havumaki, Jon Zelner.

**Funding acquisition:** Joshua Havumaki, Ted Cohen, Jon Zelner.

**Investigation:** Joshua Havumaki, Ted Cohen, Jon Zelner.

**Methodology:** Joshua Havumaki, Ted Cohen, Joel C. Miller, Jon Zelner.

**Software:** Joshua Havumaki, Chengwei Zhai, Jon Zelner.

**Supervision:** Joshua Havumaki, Ted Cohen, Seth D. Guikema, Marisa C. Eisenberg, Jon Zelner.

**Validation:** Joshua Havumaki, Ted Cohen, Jon Zelner.

**Visualization:** Joshua Havumaki, Ted Cohen, Jon Zelner.

**Writing – original draft:** Joshua Havumaki, Ted Cohen, Jon Zelner.

**Writing – review & editing:** Joshua Havumaki, Ted Cohen, Chengwei Zhai, Joel C. Miller, Seth D. Guikema, Marisa C. Eisenberg, Jon Zelner.

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
