## [Decision Letter · Decision Letter 0]

6 Oct 2020

Dear Dr. Havumaki,

Thank you very much for submitting your manuscript "Protective impacts of household-based tuberculosis contact tracing are robust across endemic incidence levels and community contact patterns" for consideration at PLOS Computational Biology.

As with all papers reviewed by the journal, your manuscript was reviewed by members of the editorial board and by several independent reviewers. In light of the reviews (below this email), we would like to invite the resubmission of a significantly-revised version that takes into account the reviewers' comments.

In addition to the reviewers' comments, I have two suggestions:

a) It would be good if the authors could also publish the model code (e.g. on github) to ensure reproducibility.

b) It would be interesting to use the model to explore alternative explanations for the mixed results of trials on HHCT (e.g. a more formal exploration of the hypothesis given in the discussion "other potential explanations for the mixed results of trials of HHCT, including the coverage and quality of case finding activities, probably are more responsible for variable results than the underlying epidemiological situation. " and/or other explanations)

We cannot make any decision about publication until we have seen the revised manuscript and your response to the reviewers' comments. Your revised manuscript is also likely to be sent to reviewers for further evaluation.

Sincerely,

Roger Dimitri Kouyos

Associate Editor

PLOS Computational Biology

Thomas Leitner

Deputy Editor

PLOS Computational Biology

In addition to the reviewers' comments, I have two suggestions:

a) It would be good if the authors could also publish the model code (e.g. on github) to ensure reproducibility.

b) It would be interesting to use the model to explore alternative explanations for the mixed results of trials on HHCT (e.g. a more formal exploration of the hypothesis given in the discussion "other potential explanations for the mixed results of trials of HHCT, including the coverage and quality of case finding activities, probably are more responsible for variable results than the underlying epidemiological situation. " and/or other explanations)

Reviewer's Responses to Questions

**Comments to the Authors:**

Reviewer #1: Havumaki et al developed a spatial model of TB transmission to understand the impact of Household contact tracing on transmission. The main justification for this undertaking are mixed results from RCTs (i.e., the Zamstar trial). The main finding is that in this model, HHCT is robust across all epidemiological setting, even in high transmission setting with community transmission. In contrast, community contact tracing appeared to be more sensitive to differences in the network model. The model relies on oversimplifications (adressed in the discussion) which are necessary for the analysis but impair generalization/translation into clinics.

Overall, I think it is an intriguing hypothesis that HHCT remains effective in settings of high community transmission. This is a well written manuscript. The main weakness of the paper is the fact that the authors rely solely on their in silico model.

Thus far, the authors focused mainly on high transmission settings. I think it would be very interesting to apply the model to low transmission settings in a more detailed way. A considerable effort is being undertaken by authorities in low transmission countries to track mostly imported cases both in the house hold contact setting as well as the community. To me, it would be very interesting to use the model to see how effective HHCT and the other tracking methods are (for example in expat communities or refugee camps) under the assumption that a certain number of new cases is imported on a regular basis from highly mobile individuals.

I don`t have the enough networking modeling experience to judge whether the methods employed are adequate and I recommend statistical review.

Reviewer #2: This article considers the transmission epidemiological features that might be expected to influence the relative impact of household contact tracing and preventive therapy on tuberculosis incidence. Their main conclusion is that the impact of this strategy is robust across epidemic intensities, which is interesting and important in my view.

Given the nature of this result, and the stochasticity involved in results and number of things varying, the main point I would seek reassurance around it that the experimental design used is adequate in terms of: a) choice of metrics used to characterise epidemiology; b) power to detect trends in outcomes with respect to these metrics. Separately, I thought it strange not to quantify the impact achieved via early detection of cases vs PT, and whether this varied by intervention strategy.

I appreciate these data may have been computationally onerous to generate (although this is not reported on), but I hope some of the below might be possible to explore by analysing the data that already exist.

Since there are three sources of variation in outputs (parameters, networks, intrinsic stochasticity), I would have found it useful to get a sense of which contributed most to overall variation, and how large the variation in impact was. The authors used the 5-year incidence end-point ratio as outcome I believe; I wonder whether they explored cumulative incidence as an outcome metric (smaller mean effect but smaller stochasticity)? In terms of metrics that may correlate with impact, I wondered about other possibilities than incidence and network features. From figure S4, there seems little relation between incidence and proportion of incidence due to household transmission. I wonder whether the proportion of transmission within the household explains more variation in the impact of HHCT? Was the time-since-incidence recorded at detection recorded? I was curious whether there were differences in the mean time as a prevalent case, and the mean number of secondary infections generated by time of detection for cases found via HHCT and via other means. In a similar vein, it would be interesting to know the typical prevalence among HH vs community contacts (both active TB and LTBI).

I think the discussion of limitations is quite focussed on technical aspects within the model universe. Heterogeneity gets a mention in future work, but perhaps clustering of tuberculosis due to shared risk factors (rather than transmission) and changes in infectiousness over time should get mentions as potentially important aspects of reality for this question that were not modelled.

Implementation details. One gathers in various places that a time-step of 1 month is used, but otherwise no details of the transmission model implementation are given, or links to source code. I think there should at least be a few sentences somewhere explaining what kind of simulation approach was used, programming language used, some sense of compute effort for results.

more minor points

- 'cdr': as a parameter name and description as 'case detection rate' might be confusing given that it does not correspond to the tuberculosis conventional case detection rate (notifications over incidence in a year)

- theta: in the parameter table this is given as a life expectancy, but I think it is the reciprocal of this as used in the model

- EL progression structure: what was the rationale for including the 5 state progression? The implied fraction progressing was stated informally (but I think differently) in the supplement and text. I would find it helpful to see in the supplement the associated survival function for the progression model (or distribution of times to progression) and actual fraction progressing (calculated for a cohort; not from the agent based model). Perhaps this could be done along with calculating mean number of secondary infections upon detection.

- Figure S14: why does this begin above 1?

- 'equilibrium': to be pedantic, the only equilibrium for a stochastic model like this is the disease free one after extinction.

- outlier exclusion: The figures contain a sentence (although it doesn't have a verb) beginning 'Among..' about excluding the 5% most extreme results. I think in the supplement somewhere this is described as outlier exclusion. First, I think this needs to be in the main article methods (sorry if I missed). Secondly, could more be presented about how much difference this makes and what is going on with the excluded runs?

- ACF: I think this makes first appearance in a section heading without previously being defined?

- spatial extent: I don't remember reading the dimensions of the spatial extent. I suppose it is implied by the houshold density; but could you state it please in methods? This would make it easier to interpret the average connection radius parameter (sigma in table 1).

Reviewer #3: Major Comments:

1. I found the description of the model a bit hard to follow, especially the spatial contact network part. Since the model is quite complex, and tries to bring different ideas and modeling techniques, the authors could be more thorough and careful with their description of the model.

See some specific comments below.

2. The results/findings are underwhelming and difficult to make sense, especially given that authors have put together a complex model to explore these phenomena.

E.g., the authors find that effect of HHCT is on average similar across low and high-incidence settings. However, low and high incidence settings can be conceived with various parameter combinations (as seen in Fig S-6), given a flexible model structure.

Hence, the relationship would really only reflect the conceptualization of low and high incidence settings, and it is unclear why one should expect the impact to vary significantly.

Perhaps focusing on specific conceptualization of the models (e.g. low incidence, with 20% HH transmission vs 50% HH transmission, or low and high incidence settings with 20% HH transmission), would help

illustrate how impact of HHCT could vary/not vary by the incidence of the setting, or by the relative role of community transmission. As it stands, it is hard to know what to make of the results because they aggregate over various scenarios.

Also, I am not convinced with the authors' conclusion that CCT are different from HHCT (see below).

Minor Comments:

Abstract:

Not sure if I follow the logic, why should individuals preferentially mix more locally in low burden settings compared to higher burden setting?

Introduction:

lines 28-29: Perhaps useful to elaborate what authors mean by social mixing patterns.

Also, in the following sentence "..longer distance between community contacts", it is unclear what distance the authors are referring to, physical distance vs distance in the contact network.

Methods:

lines 39-41: It would be helpful for the readers if some of the key features of the two previously published models that the authors are extending in this analysis are elaborated.

Household size is fixed at 5 for all households.

line 48: I think it would be useful to describe terms such as contacts, average connection radius in the context of TB transmission dynamics. eg, should a contact thought of as transmission link, potential transmission link,

or just individuals that come in to contact for some duration/frequency?

Line 60: Describe, and provide equation for global clustering coefficient, C. It is unclear how exactly clustering coefficient is calculated.

Table 2: I find it hard to believe that variance in HH transmission is the primary driver of difference in TB incidence. Differences in progression during latency, time to diagnosis, and transmission in community are important.

Results:

lines 162-163: The ratio of infection in the community to HH in the simulations range between 0.5-3, but studies find only 10-20% of infections occur in HH (ref 30). It may make sense to

explore a range that is closer to these data.

Fig 4: The correlation between CCT and average degree is to be expected.

For a given level of incidence, lower average degree would imply higher transmission between community contacts, all else being equal.

Which in turn would mean that screening fixed number of those contacts are more effective in detecting TB infection and disease.

However, the CCT does not differ much by incidence, Fig S-13 (top-left), same as for HHCT.

Hence, I am not sure that HHCT and CCT differ much by setting. I would also suggest the authors to include Fig S-13 instead of 4.

**Have all data underlying the figures and results presented in the manuscript been provided?**

Reviewer #1: Yes

Reviewer #2: **No: **Unless I missed a supplementary file, it would not be possible to reproduce the figures ie I don't think the underlying data provided).

Reviewer #3: None

PLOS authors have the option to publish the peer review history of their article (what does this mean?). If published, this will include your full peer review and any attached files.

Reviewer #1: No

Reviewer #2: No

Reviewer #3: No
---

## [Decision Letter · Decision Letter 1]

4 Jan 2021

Dear Dr. Havumaki,

Thank you very much for submitting your manuscript "Protective impacts of household-based tuberculosis contact tracing are robust across endemic incidence levels and community contact patterns" for consideration at PLOS Computational Biology. As with all papers reviewed by the journal, your manuscript was reviewed by members of the editorial board and by several independent reviewers. The reviewers appreciated the attention to an important topic. Based on the reviews, we are likely to accept this manuscript for publication, providing that you modify the manuscript according to the review recommendations.

Sincerely,

Roger Dimitri Kouyos

Associate Editor

PLOS Computational Biology

Thomas Leitner

Deputy Editor

PLOS Computational Biology

[LINK]

Reviewer's Responses to Questions

**Comments to the Authors:**

Reviewer #1: I think the authors improved the manuscript significantly.

Reviewer #2: I have no further comments.

Reviewer #4: The authors have answered all of the reviewers answers and questions. They also added some material in the manuscript and in the SI, and the additional results and sensitivity analyses are consistent with the main results. As highlighted by R1, the main weakness of the paper is the lack of application to real data. I have some additional remarks in that sense:

- the authors focused their analysis on moderate and high incidence settings (to explore the effect of HHCT in these specific cases). I think it would be useful to give some incidence values in some moderate/high incidence areas, to have an idea of where those areas/countries would compare to the model simulations

- I understand that a good amount of sensitivity analyses were already performed, but I wonder to which extend considering a more heterogeneous repartition of the population (variable number of individuals per househould, not a uniform repartition on the grid) would change (or not) the results.

- line 127: the authors constrained the rate at which individual with active TB seek care (average of 1 year). How realistic is this 1 year value?

- the authors found that HHCT resulted in lower prevalence of active TB among both households and community contacts, but the prevalence of LTBI was similar between HHCT and community CT. How can we explain that the prevalence of LTBI was not reduced by the HHCT, and which parameter in the model (i.e., which additional intervention in the HHCT) could help reducing this prevalence?

**Have all data underlying the figures and results presented in the manuscript been provided?**

Reviewer #1: Yes

Reviewer #2: Yes

Reviewer #4: Yes

PLOS authors have the option to publish the peer review history of their article (what does this mean?). If published, this will include your full peer review and any attached files.

Reviewer #1: No

Reviewer #2: No

Reviewer #4: No
---

## [Editor Report · Decision Letter 2]

14 Jan 2021

Dear Dr. Havumaki,

We are pleased to inform you that your manuscript 'Protective impacts of household-based tuberculosis contact tracing are robust across endemic incidence levels and community contact patterns' has been provisionally accepted for publication in PLOS Computational Biology.

Best regards,

Roger Dimitri Kouyos

Associate Editor

PLOS Computational Biology

Thomas Leitner

Deputy Editor

PLOS Computational Biology

---

## [Editor Report · Acceptance letter]

2 Feb 2021

PCOMPBIOL-D-20-00727R2 

Protective impacts of household-based tuberculosis contact tracing are robust across endemic incidence levels and community contact patterns

Dear Dr Havumaki,

I am pleased to inform you that your manuscript has been formally accepted for publication in PLOS Computational Biology. Your manuscript is now with our production department and you will be notified of the publication date in due course.

With kind regards,

Alice Ellingham
